# Indoxyl Sulfate Administration during Pregnancy Contributes to Renal Injury and Increased Blood–Brain Barrier Permeability

**DOI:** 10.3390/ijms241511968

**Published:** 2023-07-26

**Authors:** Ashley Griffin, Brittany Berry, Shauna-Kay Spencer, Teylor Bowles, Kedra Wallace

**Affiliations:** 1Program in Neuroscience, University of Mississippi Medical Center, 2500 North State Street, Jackson, MS 39216, USA; agriffin4@umc.edu; 2Department of Obstetrics and Gynecology, University of Mississippi Medical Center, 2500 North State Street, Jackson, MS 39216, USA; bberry@umc.edu; 3Department of Pharmacology and Toxicology, University of Mississippi Medical Center, 2500 North State Street, Jackson, MS 39216, USA; sspencer2@umc.edu (S.-K.S.); teylorbowles@yahoo.com (T.B.)

**Keywords:** blood–brain barrier, acute kidney injury, indoxyl sulfate

## Abstract

Rates of pregnancy-related acute kidney injury (PR-AKI) have increased in the U.S over the past two decades, but how PR-AKI affects the blood–brain barrier (BBB) is understudied. AKI is associated with increased amounts of uremic toxins, like indoxyl sulfate (I.S), whose chronic administration leads to BBB and cognitive changes. This study’s objective was to determine if (1) PR-AKI increases I.S and (2) if administration of I.S during pregnancy elicits renal injury and/or increases BBB permeability. From gestational day (GD) 11 to GD19, Sprague Dawley rats were given either 100 or 200 mg/kg body-weight dose of I.S. PR-AKI was induced on GD18 via 45 min bilateral renal ischemic reperfusion surgery. On GD18, metabolic cage metrics and metabolic waste was collected and on GD19 blood pressure, and BBB permeability (by Evan’s Blue infusion) were measured. I.S and creatinine were measured in both urine and circulation, respectively. One-way ANOVA or student t-tests were performed using GraphPad Prism with a *p* < 0.05 significance. I.S and PR-AKI led to oliguria. I.S administration led to increased BBB permeability compared to normal pregnant and PR-AKI animals. These results suggest that I.S administration during pregnancy leads to increased BBB permeability and evidence of renal injury comparable to PR-AKI animals.

## 1. Introduction

Pregnancy-related acute kidney injury (PR-AKI) is associated with both maternal and fetal morbidity and mortality [1]. PR-AKI rates have increased from 2.4 to 6.3 per 10,000 deliveries in the U.S from 1999 to 2011 [2]. Additionally, PR-AKI has an overall rate of hospitalizations at 0.08% in the U.S from 2006 to 2015 [3]. It is believed that PR-AKI arises from several factors, including hypertensive pregnancies and sepsis [4]. Even though PR-AKI itself remains understudied, it is well established that acute kidney injury (AKI) along with chronic kidney disease (CKD), independent of pregnancy, is associated with changes in blood–brain barrier (BBB) permeability as well as cognitive changes associated with BBB damage [5,6].

Indoxyl sulfate (I.S.), a uremic toxin, is involved in the progression from AKI to CKD as its concentration increases due to reductions in renal function [7,8]. Clinical and experimental studies have shown positive correlations between the severity of renal injury and increasing concentrations of circulating I.S. [9,10,11]. Additionally, I.S. is associated with cognitive deficits in the presence of renal injury [12,13]. We have reported that following PR-AKI, post-partum rats experience progression of renal injury and incidentally have an increase in circulating I.S. compared to control rats [14]. The objective of the current study was to determine (1) if pregnant rats with AKI had increased I.S. and (2) if administration of I.S. during pregnancy elicits renal injury and/or increases BBB permeability. We hypothesize that PR-AKI will lead to an increase in I.S. and potentially BBB permeability that is mirrored in pregnant rats administered I.S. We found that administration of I.S during pregnancy leads to renal dysfunction similar to that seen in PR-AKI rats while also increasing BBB permeability.

## 2. Results

### 2.1. Indoxyl Sulfate Excretion Is Increased following PR-AKI

To determine if experimental AKI (generated by bilateral renal ischemic reperfusion surgery on gestational day (GD) 18) led to increased I.S. and if administration of I.S. during pregnancy was associated with renal damage we measured a number of urinary biomarkers. Concentrations of urinary I.S. were significantly increased in response to PR-AKI and a higher dose of I.S. compared to normal pregnant (NP) rats (*p* = 0.003, *p* = 0.01; Figure 1A). Urinary I.S. levels were significantly lower in rats administered low doses of I.S. (100 mg/kg/day) compared to PR-AKI rats (*p* = 0.04). However, there was a significant decrease in urine output among PR-AKI (*p* = 0.007), 100 I.S. (*p* = 0.03), and 200 I.S. (*p* = 0.03) groups relative to NP rats (*p* = 0.008; Figure 1B). PR-AKI rats had significantly more circulating serum creatinine compared to NP (*p* = 0.003), 100 I.S. (*p* = 0.02), and 200 I.S. (*p* = 0.004; Figure 1C) rats. When measuring proteinuria, 200 mg/kg of I.S. led to increased proteinuria compared to NP (*p* = 0.05) and PR-AKI rats (*p* = 0.04; Figure 1D). These results suggest that while PR-AKI did not decrease I.S. excretion during pregnancy, both PR-AKI and rats administered I.S. had evidence of renal injury.

### 2.2. Indoxyl Sulfate Administration Decreases Body Weight but Not Kidney Weight during Pregnancy

The effects of I.S. administration during pregnancy on blood pressure and body and organ weight was investigated. Blood pressure was assessed at GD19, and there were no significant differences in systolic (*p* = 0.86), diastolic (*p* = 0.89), or mean arterial blood pressures (*p* = 0.85) between groups (Table 1). When accounting for bodyweight (BW), renal injury whether by PR-AKI (*p* = 0.03) or I.S. (100 mg/kg I.S. *p* = 0.009; 200 mg/kg I.S. *p* = 0.0004) administration resulted in a decrease in BW compared to NP rats. As swollen or atrophied kidneys can occur with renal injury, we determined the kidney/BW ratio among groups. PR-AKI rats had a significantly increased kidney/BW ratio relative to NP (*p* < 0.0001), 100 mg/kg I.S. (*p* = 0.003), and 200 mg/kg I.S. (*p* = 0.03) animals. These findings indicate that I.S. administration during pregnancy does not affect blood pressure but does negatively affect maternal BW.

### 2.3. Indoxyl Sulfate Administration Increases Pup Resorption but Not Placental Efficiency

As the effects of I.S. administration during pregnancy are unknown, we also assessed the litters between groups to see if there were any differences in birth outcomes. Renal injury did not significantly reduce pup weight relative to NP rats; however, pups born to AKI dams were significantly smaller than those born to dams administered 200 mg/kg I.S. (*p* = 0.003) as shown in Table 1. It should be noted that litter size, the number of pup:placenta pairs extracted from the uterus per dam, was decreased between groups (from 13.4 live pups in NP rats to 10.2 live pups in NP + 200 I.S. rats), and there was a significant increase in pup resorptions (*p* = 0.01; Figure 2) with dams administered I.S. averaging 2.5% resorptions relative to NP at 0.8% and PR-AKI at 0.5%.

To further explore the effect of PR-AKI and I.S. administration on pup in utero development, we examined placental efficiency (pup weight (g)/placenta weight(g)). Even though the decreased pup weight in PR-AKI rats did not meet statistical significance, there was a significant decrease in the placental efficiency of PR-AKI rats (3.31 ± 0.24 g) compared to the 100 mg/kg I.S. (4.19 ± 0.26 g, *p* = 0.02) and 200 mg/kg I.S. (4.24 ± 0.26 g, *p* = 0.02) groups.

### 2.4. Administration of 200 mg/kg Indoxyl Sulfate during Pregnancy Leads to an Increase in BBB Permeability

BBB permeability was assessed via infusion of Evan’s Blue, which was measured in four regions of the brain: frontal cortex, posterior cortex, brainstem, and cerebellum. PR-AKI did not significantly increase BBB permeability in the frontal cortex region of the brain (Figure 3A). Rats administered 100 mg/kg I.S. had increased permeability relative to both NP (*p* = 0.0002) and PR-AKI (*p* = 0.0003) rats in the brainstem; however, there was no change in permeability in the posterior cortex or cerebellum. BBB permeability was significantly increased in rats administered 200 mg/kg I.S. compared to NP and PR-AKI rats in the posterior cortex (*p* < 0.0001, *p* < 0.0001), brainstem (*p* = 0.002, *p* = 0.002), and cerebellum (*p* = 0.01, *p* = 0.02; Figure 3B–D), respectively. This was also true in the posterior cortex relative to rats administered 100 mg/kg I.S. (*p* < 0.0001).

## 3. Discussion

PR-AKI has been reported to be associated with an increased risk of cardiovascular events, mortality, and adverse renal outcomes; however, there is much to understand about PR-AKI’s short- and long-term effects on the central nervous system and the brain [3,15]. Clinical and rodent studies have indicated that AKI in a non-pregnant state is associated with increased risk of CKD, and the progression of AKI to CKD is associated with increased circulating I.S. [16,17]. Similar to CKD, AKI has been reported to serve as a risk factor for neuroinflammation and dementia. Clinically, AKI and CKD have been found to be associated with cognitive decline [18,19]. In male mice, a study showed severe AKI (bilateral nephrectomy or ischemic reperfusion for 60 min) led to increased BBB permeability [6]. In CKD, studies have supported an association with BBB dysfunction, which has also been associated with increased concentrations of I.S. [13]. The incidence of PR-AKI is continuing to increase worldwide and up to 14.7% of women affected by this progresses to CKD, prompting the need for studies, such as the current one [4,20]. The results of the current study demonstrate that 24 h of PR-AKI does increase I.S. but does not increase BBB permeability, whereas direct I.S. administration during pregnancy led to increases in BBB permeability. These results suggest that longer periods of PR-AKI, and the continued dysregulation of I.S., might lead to changes in BBB permeability.

BBB permeability was not increased in PR-AKI animals when assessed 24 h following injury. Similar findings have been reported in a non-pregnant animal model of ischemia/reperfusion, as tight junction proteins claudin-5 and occludin were found to be unchanged in the brain capillaries of mice subjected to kidney ischemia/reperfusion [21]. When a different non-pregnant animal model of renal injury (both the renal artery and veins were occluded for 1 h as opposed to just the renal artery) was evaluated, there were changes in hippocampal BBB function along with cognitive impairment [22]. Rats administered 200 mg/kg I.S. had significant increases in BBB permeability relative to both NP and PR-AKI rats. Previous in vitro studies have reported that I.S. increases the expression of drug transporters at both the gene and protein level at the BBB, indicating another mechanism by which I.S. may mediate neurological damage [21]. As it is known that I.S. also disrupts the BBB via activation of the aryl hydrocarbon receptor (AhR), future studies will examine BBB permeability in post-partum rats with a history of PR-AKI and examine AhR expression. Given that only rats administered high doses of I.S. had increased BBB permeability, as we believe that I.S. mediates BBB damage, it stands to reason that rats with less exposure to I.S. would not have an increase in BBB permeability. Future studies will help determine the role of I.S. in increasing BBB permeability.

Despite studies suggesting that PR-AKI and I.S. contribute to cardiovascular dysfunction and hypertension, there were no significant differences in blood pressure in the current study. We have previously published that experimental PR-AKI does not increase blood pressure when measured 24 h after injury, which is again confirmed with the current study [23]. Urine output was decreased in response to both PR-AKI and I.S. administration. This decrease in urine output combined with the increase in serum creatinine seen in PR-AKI rats is indicative of renal injury. Rats administered I.S. did not have any significant changes in creatinine but did have significantly increased proteinuria, which indicates renal damage. Studies have shown that chronic administration (4 weeks) of I.S. decreases glomerular function which would negatively impact creatinine levels [24]. Unpublished data from our lab indicates that rats administered I.S. have no changes in glomerular function, which supports the lack of change in creatinine levels when compared to NP rats. Urinary excretion of I.S. was increased in PR-AKI and 200 mg/kg I.S. rats relative to NP rats. I.S. excretion is mediated via renal organic anion transporter 1 (OAT1), which gradually loses gene expression as kidney injury progresses [25]. However, during acute cases of renal injury several animal models have reported an increase in OAT1 expression [26,27,28]. If indeed renal OAT1 expression is increased in these models similar to those previously reported, it would explain the increases in excretion of I.S. Given the high affinity of circulating I.S. to albumin, several investigators utilize mass spectrometry to measure I.S. Indeed, we are currently working on completing this assay now so that in the future we will be able to ascertain circulating levels of I.S.

Circulating levels of I.S. are inversely related to kidney function in individuals with CKD [29,30]. Due to the tight binding of I.S. to albumin in the circulation, it is hard to remove from the circulation via dialysis techniques, which ultimately contributes to the high levels that remain in the body. Following this accumulation, I.S. works directly (on vascular smooth muscle cells and endothelial cells) and indirectly (via increased expression of tissue inhibitors and growth factors) to damage organs and physiological systems [31]. Individuals with CKD are at an increased risk for cognitive impairment; a risk that has been found to be associated with increased I.S. [32]. One way that I.S. mediates its effects is through binding the AhR, which is widely expressed throughout the body, including the central nervous system. I.S. activation of AhR leads to disruption of the BBB in an animal model of renal injury [13]. Most recently patients with Alzheimer’s disease have been reported to have increased circulating levels of I.S. and disruption of AhR signaling [33], giving further evidence for the role of I.S. in contributing to cognitive dysfunction. Interestingly, the active charcoal adsorbent AST-120 adsorbs I.S., thereby reducing circulating levels of I.S. and decreasing the decline in renal function [34,35]. In addition to preventing the progression of CKD, AST-120 in an animal model of CKD also recovered cognitive function [12,36].

The current study is one of the first to evaluate the impact of I.S. administration during pregnancy. I.S. is a naturally occurring uremic toxin that occurs as a byproduct of normal digestive metabolism and has low circulating levels among healthy individuals. An ex vivo study evaluating the transfer characteristics of I.S. across the human placenta reported that there was a low maternal-to-fetal transfer of I.S. in healthy placenta cotyledons when tissues was exposed to I.S. for a short period of time (180 min) [37], further suggesting that physiologic levels of I.S. may not impair the healthy placenta. Rats administered I.S. had an increase in pup resorptions that contributed to a decrease in litter size. While pup weight was not significantly decreased, this is thought to be due in part to the reduction in litter size. No further evaluation was performed on the fetuses to see if there were any changes in development. A study by Furukawa et al. reported that fetuses from mice exposed to indole-3-acetic-acid, another tryptophan derived uremic toxin, have microencephaly and decreased neuronal formation, indicating that increased uremic toxins during pregnancy could potentially have negative effects on offspring [38]. Another study reported that offspring born to dams with experimental CKD prior to pregnancy developed hypertension [39]. As this was to our knowledge the first study to administer I.S. during pregnancy, we used doses of I.S. that have previously been reported [40]. Given the increasing rates of AKI and CKD during pregnancy and the buildup of uremic toxins that occur as a result of renal injury, we felt it was important to assess the impact of I.S. on pup outcomes at birth.

The results of the current study, taken together with those from other studies, indicate the need for further investigations into the long-term effects of PR-AKI and the role of I.S. in mediating BBB function.

## 4. Materials and Methods

### 4.1. Animal Group, Surgery, and Maintenance

Timed-pregnant Sprague Dawley rats arrived from Charles River (Boston, MA, USA) on GD10. Animals were housed in a temperature-controlled room with a 12:12 reverse light: dark cycle. All experiments were conducted in accordance with the National Institutes of Health guidelines and were approved by the Institutional Animal Care and Use Committee under protocol number 2022-1198 at the University of Mississippi Medical Center.

On GD11, animals were randomly divided into 4 experimental groups (normal pregnant (NP, *n* = 18), PR-AKI (*n* = 14), NP + 100 I.S. (*n* = 13), and NP + 200 I.S. (*n* = 12)). Rats assigned to the I.S. treatment groups began I.S. treatment (Sigma Aldrich, St Louis, MO, USA) at either 100 mg/kg BW or 200 mg/kg BW via drinking water from GD11 to 19. To our knowledge, I.S. has not previously been administered to pregnant rats before, and only one published study used females; however, most studies use either 100 or 200 mg/kg doses. For that reason, we elected to use both the 100 and 200 mg/kg dose in our study [36,40,41].

On GD18, animals underwent a 45-min bilateral renal ischemia reperfusion surgery to induce AKI as described previously [23]. Following the induction of AKI, all rats, regardless of their grouping, were placed in metabolic cages overnight with water and standard rodent chow (0.4% sodium) to allow continuous urine collection. On GD19, animals were placed in restrainers for at least 15 min on a tail warmer plate and allowed to acclimate to their environment. After acclimation, blood pressure was measured via tail cuff using the CODA noninvasive blood pressure system (Kent Scientific, Torrington, CT, USA). Following blood pressure collection, animals were either subjected to Evan’s Blue or euthanized and kidney weights, serum, and pup weights were collected. All serum and metabolic cage urine were stored at −20 °C until further analysis.

### 4.2. Assessment of BBB Permeability

A total of 4% Evan’s Blue (Sigma Aldrich, St. Louis, MO, USA) solution made in 0.9% NaCl (saline; Baxter, Deerfield, IL, USA) was infused into the jugular vein of an anesthetized rat and circulated for 30 min followed by a saline flush. The brain was dissected into the following four regions: frontal cortex (region above the anterior of the circle of Willis), posterior cortex, brainstem, and cerebellum. These four regions were selected as patients with CKD had evidence of impairment in these regions [42,43,44]. The brain regions were weighed and homogenized in trichloroacetic acid (TCA; Fisher Scientific, Fair Jawn, NJ, USA) buffer solution (1:1 20% *w*/*v* TCA and 0.9% saline) at a 1:3 dilution (*w*/*v*) as previously described [45]. The homogenized samples were centrifuged for 20 min at 10,000× *g*. The supernatant was assayed in triplicates, and samples and standards received 95% ethanol. The Evan’s Blue concentration was measured immediately at an excitation of 620 nm and an emission of 720 nm using Synergy LX (BioTek, Santa Clara, CA, USA) and Gen5 3.05 software (Biotek, Santa Clara, CA, USA). Data are presented as concentration of Evan’s Blue (pg)/weight of tissue (g).

### 4.3. Assessment of Renal Function

Using metabolic cage collections (urine, time in/out, volume of urine), urine output and urinary I.S. were measured. Urine output was calculated as total urine collected divided by time (normalized to day) spent in metabolic cage to determine oliguria. Urinary I.S concentrations (ng/mL) were measured via enzyme-linked immunosorbent assay (ELISA) (My BioSource, San Diego, CA, USA). Serum creatinine concentration (mg/dL) was determined via an assay (BioAssay Systems, Hayward, CA, USA). All samples for assays and ELISAs were ran in duplicates and per manufacturer’s instruction.

### 4.4. Statistical Analysis and Sample Size Determination

Using GraphPad Prism 9, physiological outcomes and Evan’s Blue concentrations were analyzed using a one-way ANOVA followed by Tukey’s post hoc analysis. Live and resorption pup data was analyzed by two-way ANOVA followed by Tukey’s post hoc analysis. Student t-tests were used to analyze urine output, urinary I.S, and serum creatinine. As some sample sizes were small, we assessed for normal distribution using the Shapiro–Wilk and Kolmogorov–Smimov test to ensure parametric analyses could be used. All results passed both tests of normality. Statistical significance was determined as *p* < 0.05, and data were represented as mean ± standard error mean.

Using a family-wise alpha level of 0.05, and assuming an effect size of Cohen’s d = 2.44 (based on data from our original PR-AKI study [23]), our sample size provides over 85% power to compare each of the experimental groups with the NP control groups.

## Figures and Tables

**Figure 1 ijms-24-11968-f001:**
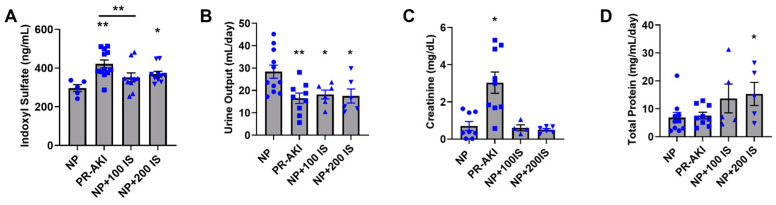
Urinary indoxyl sulfate (I.S.) and kidney assessment in late pregnancy: (**A**) Urinary I.S. was increased in PR-AKI and NP + 200 I.S. rats compared to NP rats, and PR-AKI was increased vs. NP + 100 I.S.; (**B**) urine excretion collected by metabolic cages was significantly decreased in PR-AKI and I.S. rats vs. NP rats; (**C**) serum creatinine was increased in PR-AKI rats vs. all other groups. (**D**) total urinary protein was increased in NP + 200 I.S. rats compared to NP and PR-AKI rats. Individual data points are represented as follows: NP (circles), PR-AKI (squares), NP + 100 I.S. (upward triangle), and NP + 200 I.S. (downward triangle). * *p* < 0.05, ** *p* < 0.005.

**Figure 2 ijms-24-11968-f002:**
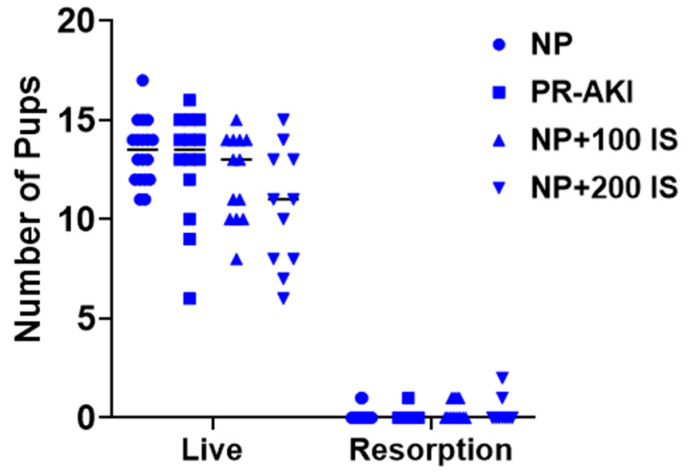
Number of live and resorption pups. The number of live pups decreased significantly in NP + 200 I.S. compared to NP and PR-AKI animals. Pup resorption was significantly increased in NP + 200 I.S. compared to NP and PR-AKI animals. Individual data points are represented as followed: NP (circles), PR-AKI (squares), NP + 100 I.S. (upward triangle), and NP + 200 I.S. (downward triangle).

**Figure 3 ijms-24-11968-f003:**
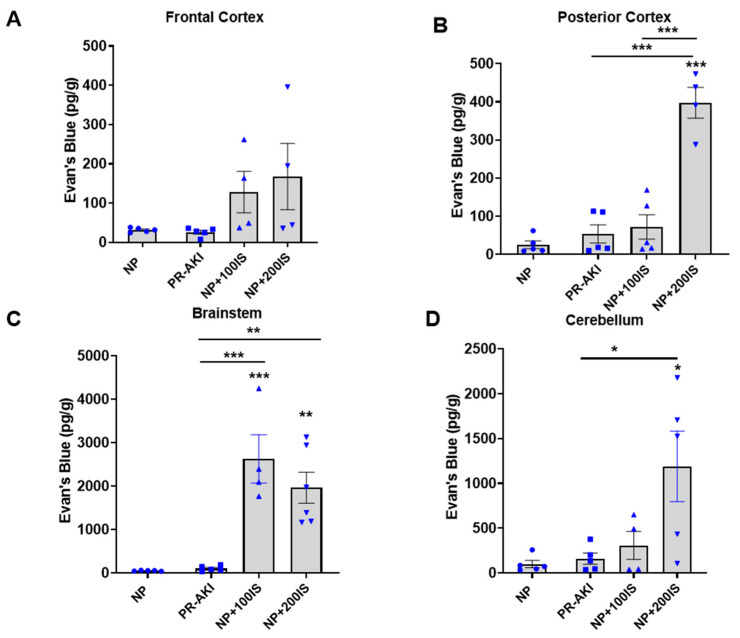
Evan’s Blue concentration in brain regions: (**A**) In the frontal cortex, there was no difference in Evan’s Blue concentration between groups. (**B**) In the posterior cortex, NP + 200 I.S. had significantly more Evan’s Blue concentration than NP, PR-AKI, and NP + 100 I.S. groups. (**C**) In the brain stem, both NP + 100 I.S. and NP + 200 I.S. had significantly more Evan’s Blue concentration than NP and PR-AKI groups. (**D**) In the cerebellum, NP + 200 I.S. had significantly more Evan’s Blue concentration compared to NP and PR-AKI groups. Individual data points are represented as followed: NP (circles), PR-AKI (squares), NP + 100 I.S. (upward triangle), and NP + 200 I.S. (downward triangle). * *p* < 0.05, ** *p* < 0.005, *** *p* < 0.0005.

**Table 1 ijms-24-11968-t001:** Physiological outcomes measured at GD19 of pregnancy.

	NP	PR-AKI	NP + 100 I.S.	NP + 200 I.S.	*p* Value
Systolic Pressure (mmHg)	126.2 ± 12.52	124.3 ± 3.37	129.9 ± 7.02	124.7 ± 4.68	0.86
Diastolic Pressure (mmHg)	91.3 ± 3.04	89.5 ± 2.63	94.0 ± 5.78	90.7 ± 4.1	0.89
Mean Arterial Pressure (mmHg)	102.7 ± 3.29	100.6 ± 2.82	105.9 ± 6.14	101.5 ± 4.23	0.85
Body Weight (g)	338.4 ± 8.3	308.1 ± 10.68 *	305.8 ± 7.29 **	288.3 ± 9.08 ***	0.002
Kidney/Body Weight (g/g)	2.8 ± 0.12 ^+++^	3.6 ± 0.12	3.0 ± 0.14 ^++^	3.1 ± 0.11 ^+^	<0.0001
Pup Weight (g)	2.1 ± 0.15	1.7 ± 0.15	2.1 ± 0.14	2.5 ± 0.14 ^++^	0.007

* Denotes *p* < 0.05, ** *p* < 0.005 compared to NP, *** *p* < 0.0001 compared to NP; ^+^ denotes *p* < 0.05, ^++^
*p* < 0.005, ^+++^
*p* < 0.0001 compared to NP + AKI.

## Data Availability

Data are contained within this article.

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
