# Peer review of "Indoxyl Sulfate Administration during Pregnancy Contributes to Renal Injury and Increased Blood–Brain Barrier Permeability"

_ijms, 2023, doi:10.3390/ijms241511968_

Round 1

Reviewer 1 Report

The authors describe well and conceptualize through the use of graphs and tables. In particular the caption of the figures are a good guide for the eye and make the focus of the authors easier. Personally I would have made the discussions longer and a few more references .Accept in present form

Author Response

Comments: The authors describe well and conceptualize through the use of graphs and tables. In particular the caption of the figures are a good guide for the eye and make the focus of the authors easier. Personally I would have made the discussions longer and a few more references .Accept in present form

Response: We thank the reviewer for their comments, and in our revised manuscript we have extended our discussion and added additional references. In our revised manuscript we have worked to make our results clearer. 

Reviewer 2 Report

Griffin et al deal with an interesting issue. My major concerns are:

1. The number of the animals included has to be clarified. In case that each group has less than 20, non parametric tests have to performed. In the same case, they also have to to correct the bias of small samples (Hedge's g or ssev.eu).

2. Forest plots have to replace the charts presented. This will increase the credibility of the results of this good work.

3. the bioethics issues that  this study supports have to further discussed

Author Response

Comment 1: Griffin et al deal with an interesting issue. My major concerns are: The number of the animals included has to be clarified. In case that each group has less than 20, non parametric tests have to performed. In the same case, they also have to correct the bias of small samples (Hedge's g or ssev.eu).

Response 1: We thank the reviewer for this suggestion and submitted our work for review by a biostatistician. We applied the Shapiro-Wilk and the Kolmogorov-Smirnov test to assess normal distribution for our assays. All tests passed the normal distribution assumption so it was recommended that we leave the analysis as is. However, as the reviewer pointed out our sample sizes are small so we included these additional analyses and sample size results into the revised manuscript.

Comment 2: Forest plots have to replace the charts presented. This will increase the credibility of the results of this good work.

Response 2: We thank the reviewer for the suggestion, however after having our statistics reviewed it was not recommended as we are not presenting any data from or for a systematic literature review.

Comment 3: the bioethics issues that this study supports have to further discussed

Response 3: We appreciate this comment by the reviewer. In our revised manuscript we have ensured that our IACUC information is in the statement section. In the discussion we have added information regarding how I.S. is a naturally occurring uremic toxin that is in everyone and an additional human study showing that the maternal-to-fetal- transfer of I.S. (using an ex vivo system) is extremely low in a healthy pregnancy.

Reviewer 3 Report

In this manuscript, the authors adopted the PR-AKI model and Indoxyl Sulfate (I.S.) treatment, and found that I.S. treatment induced increased BBB permeability. Despite a list of interesting findings, there are several concerns that may weaken the manuscript.

1. Results section (2.1 and 2.2): please add some introductory sentence in the beginning and some conclusion sentence at the end of each subsection.

2. How the PR-AKI model is generated? I noticed the detailed description in the methods section. Need to briefly describe the mice model in the beginning of the results section.

3. PR-AKI group showed the highest level of urine I.S., the authors also collected the serum samples. I suppose the authors should measure the serum I.S. concentration as well.

4. Figure 3 showed increased permeability of BBB in the NP+100 I.S. and NP + 200 I.S. groups, but not in the PR-AKI groups. The authors mentioned in the discussion section that the PR-AKI group might be too short (only 24hr post-injury). This makes me question the suitability of the PR-AKI model in this study. Should the author perform PR-AKI on GD11 as well, the same time point when the pregnant rats were administered with I.S.?

5. Conclusion in the abstract: These results suggest that acute I.S administration leads to increased BBB permeability and evidence of renal injury comparable to PR-AKI animals. This statement is not accurate: compared to PR-AKI (24 hours), I.S. administration lasts from GD11 to GD19, which is not acute, but chronic.

6. In the results section 2.3, the authors mention fetal weight, does this result refer to the table 1 (pup weight)? Also, the authors mention little size, what does "little size" mean? Please clarify.

Overall, the quality of English language is good.

Author Response

Comment 1. Results section (2.1 and 2.2): please add some introductory sentence in the beginning and some conclusion sentence at the end of each subsection.

Response 1. We appreciate this suggestion by the reviewer and in the revised manuscript, we have integrated these suggestions.

Comment 2: How is the PR-AKI model generated? I noticed the detailed description in the methods section. Need to briefly describe the mice model in the beginning of the results section.

Response 2: We thank the reviewer for this suggestion, and in the revised manuscript we included information about the model at the beginning of the results section.

Comment 3: PR-AKI group showed the highest level of urine I.S., the authors also collected the serum samples. I suppose the authors should measure the serum I.S. concentration as well.

Response 3:  We thank the author for this comment and agree. We are currently working with the mass spectrometry core to measure serum and plasma concentrations of I.S. As there are controversial differences regarding the use of ELISAs to measure circulating I.S. and the sensitivity of the uremic toxins we have elected to use mass spectrometry. We have added this information and some more regarding the binding of albumin to I.S. in the revised discussion.

Comment 4: Figure 3 shows increased permeability of BBB in the NP+100 I.S. and NP + 200 I.S. groups, but not in the PR-AKI groups. The authors mentioned in the discussion section that the PR-AKI group might be too short (only 24hr post-injury). This makes me question the suitability of the PR-AKI model in this study. Should the author perform PR-AKI on GD11 as well, the same time point when the pregnant rats were administered with I.S.?

Response 4: We thank the reviewer for this comment and have clarified our statement and overall hypothesis in the revised discussion. In short we believe that I.S. accumulates over time to lead to the progression of kidney injury and BBB damage. As such we should not see direct damage to the BBB 24hrs post injury.

Comment 5: Conclusion in the abstract: These results suggest that acute I.S administration leads to increased BBB permeability and evidence of renal injury comparable to PR-AKI animals. This statement is not accurate: compared to PR-AKI (24 hours), I.S. administration lasts from GD11 to GD19, which is not acute, but chronic.

Response 5: We apologize for the misuse of the term and have removed the “acute” when in reference to I.S administration in the abstract.

Comment 6: In the results section 2.3, the authors mention fetal weight, does this result refer to table 1 (pup weight)? Also, the authors mention little size, what does "little size" mean? Please clarify.

Response 6: We apologize for the inconsistency in terms and yes the reviewer is correct in that fetal weight does refer to pup weight. In the revised manuscript we have made these corrections. We have also clarified that litter size refers to the number of pup:placenta pairs extracted from the uterus per dam.

Round 2

Reviewer 2 Report

Small samples bias has not been addressed, thus, the study can not be published in a such distinguished journal.

Reviewer 3 Report

The revised manuscript has improved a lot. I'm satisfied with the revised version and the author's response.